# Connected Superlevel Set in (Deep) Reinforcement Learning and its Application to Minimax Theorems

**Sihan Zeng**
Electrical and Computer Engineering
Georgia Institute of Technology
Atlanta, GA 30318
szeng30@gatech.edu

**Thinh T. Doan**
Electrical and Computer Engineering
Virginia Tech
Blacksburg, VA 24061
thinhdoan@vt.edu

**Justin Romberg**
Electrical and Computer Engineering
Georgia Institute of Technology
Atlanta, GA 30318
jrom@ece.gatech.edu

## Abstract

The aim of this paper is to improve the understanding of the optimization landscape for policy optimization problems in reinforcement learning. Specifically, we show that the superlevel set of the objective function with respect to the policy parameter is always a connected set both in the tabular setting and under policies represented by a class of neural networks. In addition, we show that the optimization objective as a function of the policy parameter and reward satisfies a stronger "equiconnectedness" property. To our best knowledge, these are novel and previously unknown discoveries.

We present an application of the connectedness of these superlevel sets to the derivation of minimax theorems for robust reinforcement learning. We show that any minimax optimization program which is convex on one side and is equiconnected on the other side observes the minimax equality (i.e. has a Nash equilibrium). We find that this exact structure is exhibited by an interesting class of robust reinforcement learning problems under an adversarial reward attack, and the validity of its minimax equality immediately follows. This is the first time such a result is established in the literature.

## 1 Introduction

Policy optimization problems in reinforcement learning (RL) are usually formulated as the maximization of a non-concave objective function over a convex constraint set. Such non-convex programs are generally difficult to solve globally, as gradient-based optimization algorithms can be trapped in sub-optimal first-order stationary points. Interestingly, recent advances in RL theory [Fazel et al., 2018, Agarwal et al., 2021, Mei et al., 2020] have discovered a "gradient domination" structure in the optimization landscape, which qualitatively means that every stationary point of the objective function is globally optimal. An important consequence of this condition is that any first-order algorithm that converges to a stationary point is guaranteed to find the global optimality.

In this work, our aim is to enhance the understanding of the optimization landscape in RL beyond the gradient domination condition. Inspired by Mohammadi et al. [2021], Fatkhullin and Polyak [2021] that discuss properties of the sublevel set for the linear-quadratic regulator (LQR), we study

37th Conference on Neural Information Processing Systems (NeurIPS 2023).

the superlevel set of the policy optimization objective under a Markov decision process (MDP) framework and prove that it is always connected.

As an immediate consequence, we show that any minimax optimization program which is convex on one side and is an RL objective on the other side observes the minimax equality. We apply this result to derive an interesting and previously unknown minimax theorem for robust RL. We also note that it is unclear at the moment, but certainly possible, that the result on connected superlevel sets may be exploited to design more efficient and reliable policy optimization algorithms in the future.

## 1.1   Main Contribution

Our first contribution in this work is to show that the superlevel set of the policy optimization problem in RL is always connected under a tabular policy representation. We then extend this result to the deep reinforcement learning setting, where the policy is represented by a class of over-parameterized neural networks. We show that the superlevel set of the underlying objective function with respect to the policy parameters (i.e. weights of the neural networks) is connected at all levels. We further prove that the policy optimization objective as a function of the policy parameter and reward is "equiconnected", which is a stronger result that we will define and introduce later in the paper. To the best of our knowledge, our paper is the first to rigorously investigate the connectedness of the superlevel sets for the MDP policy optimization program, both in the tabular case and with a neural network policy class.

As a downstream application, we discuss how our main results can be used to derive a minimax theorem for a class of robust RL problems. We consider the scenario where an adversary strategically modifies the reward function to trick the learning agent. Aware of the attack, the learning agent defends against the poisoned reward by solving a minimax optimization program. The formulation for this problem is proposed and considered in Banihashem et al. [2021], Rakhsha et al. [2020]. However, as a fundamental question, the validity of the minimax theorem (or equivalently, the existence of a Nash equilibrium) is still unknown. We fill in this gap by establishing the minimax theorem as a simple consequence of the equiconnectedness of the policy optimization objective.

## 1.2   Related Works

Our paper is closely connected to the existing works that study the structure of policy optimization problems in RL, especially those on the gradient domination condition. Our result also relates to the literature on minimax optimization for various function classes and robust RL. We discuss the recent advances in these domains to give context to our contributions.

**Gradient Domination Condition.** The policy optimization problem in RL is non-convex but obeys the special "gradient domination" structure, which has been widely used as a tool to show the convergence of various gradient-based algorithms to the globally optimal policy [Agarwal et al., 2020, Mei et al., 2020, Bhandari and Russo, 2021, Zeng et al., 2021a, Xiao, 2022]. In the settings of LQR [Fazel et al., 2018, Yang et al., 2019] and entropy-regularized MDP [Mei et al., 2020, Cen et al., 2022, Zeng et al., 2022], the gradient domination structure can be mathematically described by the Polyak-Łojasiewicz (PŁ) condition, which bears a resemblance to strong convexity but does not even imply convexity. It is known that functions observing this condition can be optimized globally and efficiently by (stochastic) optimization algorithms [Karimi et al., 2016, Zeng et al., 2021b, Gower et al., 2021]. When the policy optimization problem under a standard, non-regularized MDP is considered, the gradient domination structure is weaker than the PŁ condition but still takes the form of upper bounding a global optimality gap by a measure of the magnitude of the gradient [Bhandari and Russo, 2019, Agarwal et al., 2020, 2021]. In all scenarios, the gradient domination structure prevents any stationary point from being sub-optimal.

It may be tempting to think that the gradient domination condition and the connectedness of the superlevel sets are strongly connected notions or may even imply one another. For 1-dimensional function ($f : \mathbb{R}^n \to \mathbb{R}$ with $n = 1$), it is easy to verify that the gradient domination condition necessarily implies the connectedness of the superlevel sets. However, when $n \geq 2$ this is no longer true. In general, the gradient domination condition neither implies nor is implied by the connectedness of superlevel sets, which we illustrate with examples in Section 1.3. These two structural properties are distinct concepts that characterize the optimization landscape from different angles. This observation precludes the possibility of deriving the connectedness of the superlevel

sets in RL simply from the existing results on the gradient domination condition, and suggests that a tailored analysis is required.

**Minimax Optimization & Minimax Theorems.** Consider a function $f : \mathcal{X} \times \mathcal{Y} \to \mathbb{R}$ on convex sets $\mathcal{X}, \mathcal{Y}$. In general, the minimax inequality always holds

$$\sup_{x \in \mathcal{X}} \inf_{y \in \mathcal{Y}} f(x, y) \leq \inf_{y \in \mathcal{Y}} \sup_{x \in \mathcal{X}} f(x, y).$$

The seminal work Neumann [1928] shows that this inequality holds as an equality for matrix games where $\mathcal{X} \subseteq \mathbb{R}^m, \mathcal{Y} \subseteq \mathbb{R}^n$ are probability simplexes and we have $f(x, y) = x^\top A y$ given a payoff matrix $A \in \mathbb{R}^{m \times n}$. The result later gets generalized to the setting where $\mathcal{X}, \mathcal{Y}$ are compact sets, $f(x, \cdot)$ is quasi-convex for all $x \in \mathcal{X}$, and $f(\cdot, y)$ is quasi-concave for all $y \in \mathcal{Y}$ [Fan, 1953, Sion, 1958]. Much more recently, Yang et al. [2020] establishes the minimax equality when $f$ satisfies the two-sided PŁ condition. For arbitrary functions $f$, the minimax equality need not be valid.

The validity of the minimax equality is essentially equivalent to the existence of a global Nash equilibrium $(x^\star, y^\star)$ such that

$$f(x, y^\star) \leq f(x^\star, y^\star) \leq f(x^\star, y), \quad \forall x \in \mathcal{X}, y \in \mathcal{Y}.$$

The Nash equilibrium $(x^\star, y^\star)$ is a point where neither player can improve their objective function value by changing its strategy. In general nonconvex-nonconcave settings where the global Nash equilibrium may not exist, alternative approximate local/global optimality notions are considered [Daskalakis and Panageas, 2018, Nouiehed et al., 2019, Adolphs et al., 2019, Jin et al., 2020].

**Robust Reinforcement Learning.** Robust RL studies finding the optimal policy in the worst-case scenario under environment uncertainty and/or possible adversarial attacks. Various robust RL models have been considered in the existing literature, such as: 1) the learning agent operates under uncertainty in the transition probability kernel [Goyal and Grand-Clement, 2022, Li et al., 2022, Panaganti and Kalathil, 2022, Wang et al., 2023], 2) an adversary exists and plays a two-player zero-sum Markov game against the learning agent [Pinto et al., 2017, Tessler et al., 2019], 3) the adversary does not affect the state transition but may manipulate the state observation [Havens et al., 2018, Zhang et al., 2020], 4) there is uncertainty or attack only on the reward [Wang et al., 2020, Banihashem et al., 2021, Sarkar et al., 2022], 5) the learning agent defends against attacks from a population of adversaries rather than a single one [Vinitsky et al., 2020]. A particular attack and defense model considered later in our paper is adapted from Banihashem et al. [2021].

**Other Works on Connected Level Sets in Machine Learning.** Last but not least, we note that our paper is related to the works that study the connectedness of the sublevel sets for the LQR optimization problem [Fatkhullin and Polyak, 2021] and for deep supervised learning under a regression loss [Nguyen, 2019]. The neural network architecture considered in our paper is inspired by and similar to the one in Nguyen [2019]. However, our result and analysis on deep RL are novel and significantly more challenging to establish, since 1) the underlying loss function in Nguyen [2019] is convex, while ours is a non-convex policy optimization objective, 2) the analysis of Nguyen [2019] relies critically on the assumption that the activation functions are uniquely invertible, while we use a non-uniquely invertible softmax activation function to generate policies within the probability simplex.

### 1.3 Connection between Gradient Domination and Connected Superlevel Sets

We loosely use the term "gradient domination" to indicate that a differentiable function does not have any sub-optimal stationary points. In this section, we use two examples to show that the gradient domination condition in general does not imply or get implied by the connectedness of the superlevel sets. The first example is a function that observes the gradient domination condition but has a disconnected set of maximizers (which implies that the superlevel is not always connected).

Consider $f : [-4, 4] \times [-2, 0] \to \mathbb{R}$

$$f(x, y) = \begin{cases} f_1(x, y) = -(x-1)^3 + 3(x-1) - y^2 - 2y - 0.02(y+10)^2(10 - x^2), & \text{for } x \geq 0 \\ f_2(x, y) = -(-x-1)^3 + 3(-x-1) - y^2 - 2y - 0.02(y+10)^2(10 - x^2), & \text{else} \end{cases}$$

It is obvious that the function is symmetric along the line $x = 0$ and that $f_1(0, y) = f_2(0, y)$ for all $y \in [-2, 0]$. Computing the derivatives of $f_1$ and $f_2$ with respect to $x$, we have

$$\nabla_x f_1(x, y) = -3(x-1)^2 + 3 + 0.04x(y+10)^2,$$
$$\nabla_x f_2(x, y) = 3(x+1)^2 - 3 + 0.04x(y+10)^2.$$

We can again verify $\nabla_x f_1(0, y) = \nabla_x f_2(0, y)$ for all $y$, which implies that the function $f$ is everywhere continuous and differentiable. Visualization of $f$ in Fig. 1 along with simple calculation (solving the system of equations $\nabla_x f(x, y) = 0$ and $\nabla_y f(x, y) = 0$) show that there are only two stationary points of $f$ on $[-4, 4] \times [-2, 0]$. The two stationary points are $(3.05, -1.12)$ and $(-3.05, -1.12)$, and they are both global maximizers on this domain, which means that the gradient domination condition is observed. However, the set of maximizers $\{(3.05, -1.12), (-3.05, -1.12)\}$ is clearly disconnected.

We next present a function that has connected superlevel sets at all level but does not observe the gradient domination condition (i.e. has sub-optimal stationary points).

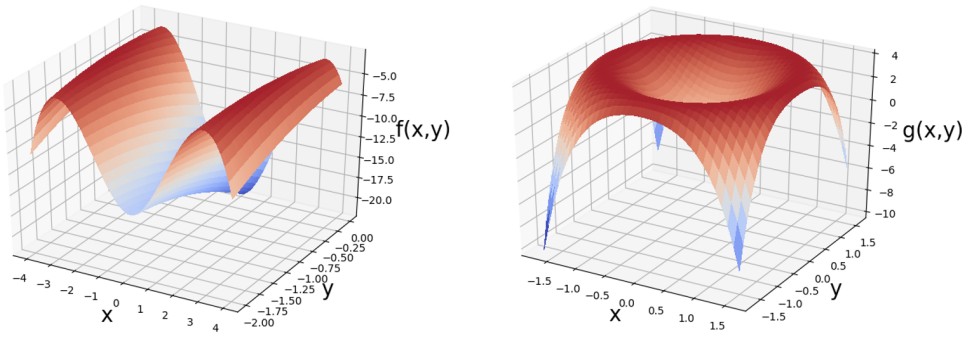

Figure 1: Visualization of Functions $f$ (Left) and $g$ (Right)

Consider $g : \mathbb{R}^2 \to \mathbb{R}$ defined as

$$g(x, y) = -(x^2 + y^2)^2 + 4(x^2 + y^2).$$

This is a volcano-shaped function, which we visualize in Fig. 1. It is obvious the superlevel set $\{(x, y) : g(x, y) \geq \lambda\}$ is always either a 2D circle (convex set) or a donut-shaped connected set depending on the choice of $\lambda$. However, the gradient domination condition does not hold as $(0, 0)$ is a first-order stationary point but not a global maximizer (it is actually a local minimizer).

**Outline of the paper.** The rest of the paper is organized as follows. In Section 2, we discuss the policy optimization problem in the tabular setting and establish the connectedness of the superlevel sets. Section 3 generalizes the result to a class of policies represented by over-parameterized neural networks. We introduce the structure of the neural network and the definition of super level sets in this context, and present our theoretical result. In Section 4, we use our main results on superlevel sets to derive two minimax theorems for robust RL. Finally, we conclude in Section 5 with remarks on future directions.

## 2  Connected Superlevel Set Under Tabular Policy

We consider the infinite-horizon average-reward MDP characterized by $\mathcal{M} = (\mathcal{S}, \mathcal{A}, \mathcal{P}, r)$. We use $\mathcal{S}$ and $\mathcal{A}$ to denote the state and action spaces, which we assume are finite. The transition probability kernel is denoted by $\mathcal{P} : \mathcal{S} \times \mathcal{A} \to \Delta_{\mathcal{S}}$, where $\Delta_{\mathcal{S}}$ denotes the probability simplex over $\mathcal{S}$. The reward function $r : \mathcal{S} \times \mathcal{A} \to [0, U_r]$ is bounded for some positive constant $U_r$ and can also be regarded as a vector in $\mathbb{R}^{|\mathcal{S}| \times |\mathcal{A}|}$. We use $P^{\pi} \in \mathbb{R}^{\mathcal{S} \times \mathcal{S}}$ to represent the state transition probability matrix under policy $\pi \in \Delta_{\mathcal{A}}^{\mathcal{S}}$, where $\Delta_{\mathcal{A}}^{\mathcal{S}}$ is the collection of probability simplexes over $\mathcal{A}$ across the state space

$$P_{s',s}^{\pi} = \sum_{a \in \mathcal{A}} \mathcal{P}(s' \mid s, a) \pi(a \mid s), \quad \forall s', s \in \mathcal{S}. \tag{1}$$

We consider the following ergodicity assumption in the rest of the paper, which is commonly made in the RL literature [Wang, 2017, Wei et al., 2020, Wu et al., 2020].

**Assumption 1** *Given any policy $\pi$, the Markov chain formed under the transition probability matrix $P^{\pi}$ is ergodic, i.e. irreducible and aperiodic.*

Let $\mu_\pi \in \Delta_\mathcal{S}$ denote the stationary distribution of the states induced by policy $\pi$. As a consequence of Assumption 1, the stationary distribution $\mu_\pi$ is unique and uniformly bounded away from 0 under any $\pi$. In addition, $\mu_\pi$ is the unique eigenvector of $P^\pi$ with the associated eigenvalue equal to 1, i.e. $\mu_\pi = P^\pi \mu_\pi$. Let $\widehat{\mu}_\pi \in \Delta_{\mathcal{S} \times \mathcal{A}}$ denote the state-action stationary distribution induced by $\pi$, which can be expressed as

$$\widehat{\mu}_\pi(s,a) = \mu_\pi(s)\pi(a \mid s). \tag{2}$$

We measure the performance of a policy $\pi$ under reward function $r$ by the average cumulative reward $J_r(\pi)$

$$J_r(\pi) \triangleq \lim_{K \to \infty} \frac{\sum_{k=0}^{K} r(s_k, a_k)}{K} = \mathbb{E}_{s \sim \mu_\pi, a \sim \pi}[r(s_k, a_k)] = \sum_{s,a} r(s,a)\widehat{\mu}_\pi(s,a).$$

The objective of the policy optimization problem is to find the policy $\pi$ that maximizes the average cumulative reward

$$\max_{\pi \in \Delta_\mathcal{A}^\mathcal{S}} J_r(\pi). \tag{3}$$

The superlevel set of $J_r$ is the set of policies that achieve a value function greater than or equal to a specified level. Formally, given $\lambda \in \mathbb{R}$, the $\lambda$-superlevel set (or superlevel set) under reward $r$ is defined as

$$\mathcal{U}_{\lambda,r} \triangleq \{\pi \in \Delta_\mathcal{A}^\mathcal{S} \mid J_r(\pi) \geq \lambda\}.$$

The main focus of this section is to study the connectedness of this set $\mathcal{U}_{\lambda,r}$, which requires us to formally define a connected set.

**Definition 1** *A set $\mathcal{U}$ is connected if for any $x,y \in \mathcal{U}$ there exists a continuous map $p : [0,1] \to \mathcal{U}$ such that $p(0) = x$ and $p(1) = y$.*

We say that a function is connected if its superlevel sets are connected at all levels. We also introduce the definition of equiconnected functions.

**Definition 2** *Given two spaces $\mathcal{X}$ and $\mathcal{Y}$, the collection of functions $\{f_y : \mathcal{X} \to \mathbb{R}\}_{y \in \mathcal{Y}}$ is said to be equiconnected if for every $x_1, x_2 \in \mathcal{X}$, there exists a continuous path map $p : [0,1] \to \mathcal{X}$ such that*

$$p(0) = x_1, \quad p(1) = x_2, \quad f_y(p(\alpha)) \geq \min\{f_y(x_1), f_y(x_2)\},$$

*for all $\alpha \in [0,1]$ and $y \in \mathcal{Y}$.*

Conceptually, the collection of functions $\{f_y : \mathcal{X} \to \mathbb{R}\}_{y \in \mathcal{Y}}$ being equiconnected requires 1) that $f_y(\cdot)$ is a connected function for all $y \in \mathcal{Y}$ (or equivalently, the set $\{x \in \mathcal{X} : f_y(x) \geq \lambda\}$ is connected for all $\lambda \in \mathbb{R}$ and $y \in \mathcal{Y}$) and 2) that the path map constructed to prove the connectedness of $\{x \in \mathcal{X} : f_y(x) \geq \lambda\}$ is independent of $y$.

We now present our first main result of the paper, which states that the superlevel set $\mathcal{U}_{\lambda,r}$ is always connected.

**Theorem 1** *Under Assumption 1, the superlevel set $\mathcal{U}_{\lambda,r}$ is connected for any $\lambda \in \mathbb{R}$ and $r \in \mathbb{R}^{|\mathcal{S}||\mathcal{A}|}$. In addition, the collection of functions $\{J_r(\cdot) : \Delta_\mathcal{A}^\mathcal{S} \to \mathbb{R}\}_{r \in \mathbb{R}^{|\mathcal{S}| \times |\mathcal{A}|}}$ is equiconnected.*

Our result here extends easily to the infinite-horizon discounted-reward setting since a discounted-reward MDP can be regarded as an average-reward one with a slightly modified transition kernel [Konda, 2002].

The claim in Theorem 1 on the equiconnectedness of $\{J_r\}_{r \in \mathbb{R}^{|\mathcal{S}| \times |\mathcal{A}|}}$ is a slightly stronger result than the connectedness of $\mathcal{U}_{\lambda,r}$, and plays an important role in the application to minimax theorems discussed later in Section 4.

We note that the proof, presented in Section A.1 of the appendix, mainly leverages the fact that the value function $J_r(\pi)$ is linear in the state-action stationary distribution $\widehat{\mu}_\pi$ and that there is a special connection (though nonlinear and nonconvex) between $\widehat{\mu}_\pi$ and the policy $\pi$, which we take advantage

of to construct the continuous path map for the analysis. Specifically, given two policies $\pi_1, \pi_2$ with $J_r(\pi_1), J_r(\pi_2) \geq \lambda$, we show that the policy $\pi_\alpha$ defined as

$$\pi_\alpha(a \mid s) = \frac{\alpha \mu_{\pi_1}(s) \pi_1(a \mid s) + (1 - \alpha) \mu_{\pi_2}(s) \pi_2(a \mid s)}{\alpha \mu_{\pi_1}(s) + (1 - \alpha) \mu_{\pi_2}(s)}, \quad \forall \alpha \in [0, 1]$$

is guaranteed to achieve $J_r(\pi_\alpha) \geq \lambda$ for all $\alpha \in [0, 1]$.

Besides playing a key role in the proof of Theorem 1, our construction of this path map may inform the design of algorithms in the future. Given any two policies with a certain guaranteed performance, we can generate a continuum of policies at least as good. As a consequence, if we find two optimal policies (possibly by gradient descent from different initializations) we can generate a range of interpolating optimal policies. If the agent has a preference over these policy (for example, to minimize certain energy like in $H_1$ control, or if some policies are easier to implement physically), then the selection can be made on the continuum of optimal policies, which eventually leads to a more preferred policy.

## 3   Connected Superlevel Set Under Neural Network Parameterized Policy

In real-world reinforcement learning applications, it is common to use a deep neural network to parameterize the policy [Silver et al., 2016, Arulkumaran et al., 2017]. In this section, we consider the policy optimization problem under a special class of policies represented by an over-parameterized neural network and show that this problem still enjoys the important structure — the connectedness of the superlevel sets — despite the presence of the highly complex function approximation. Illustrated in Fig. 2, the neural network parameterizes the policy in a very natural manner which matches how neural networks are actually used in practice.

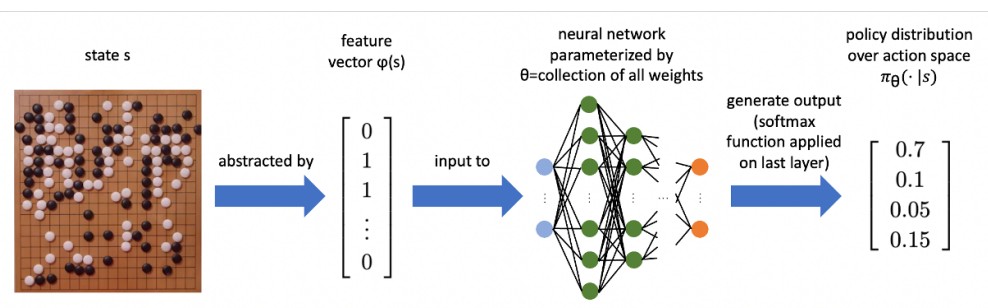

Figure 2: Neural Network Policy Representation

Mathematically, the parameterization can be described as follows. Each state $s \in \mathcal{S}$ is associated with a feature vector $\phi(s) \in \mathbb{R}^d$, which in practice is usually carefully selected to summarize the key information of the state. For state identifiability, we assume that the feature vector of each state is unique, i.e.

$$\phi(s) \neq \phi(s'), \quad \forall s, s' \in \mathcal{S} \text{ and } s \neq s'.$$

To map a feature vector $\phi(s)$ to a policy distribution over state $s$, we employ a $L$-layer neural network, which in the $k_{\text{th}}$ layer has weight matrix $W_k \in \mathbb{R}^{n_{k-1} \times n_k}$ and bias vector $b_k \in \mathbb{R}^{n_k}$ with $n_0 = d$ and $n_L = |\mathcal{A}|$. For the simplicity of notation, we use $\Omega_k$ to denote the space of weight and bias parameters $(W_k, b_k)$ of layer $k$, and we write $\Omega = \Omega_1 \times \cdots \times \Omega_L$. We use $\theta$ to denote the collection of the weights and biases

$$\theta = ((W_1, b_1), \cdots, (W_L, b_L)) \in \Omega$$

We use the same activation function for layers 1 through $L - 1$, denoted by $\sigma : \mathbb{R} \to \mathbb{R}$, applied in an element-wise fashion to vectors. To ensure that the output of the neural network is a valid probability distribution, the activation function for the last layer is a softmax function, denoted by $\psi : \mathbb{R}^{|\mathcal{A}|} \to \Delta_{\mathcal{A}}$, i.e. for any vector $v \in \mathbb{R}^{|\mathcal{A}|}$

$$\psi(v)_i = \frac{\exp(v_i)}{\sum_{i'=1}^{|\mathcal{A}|} \exp(v_{i'})}, \quad \forall i = 1, ..., |\mathcal{A}|.$$

With $v \in \mathbb{R}^d$ as the input to a neural network with parameters $\theta$, we use $f_k^\theta(v) \in \mathbb{R}^{n_k}$ to denote the output of the network at layer $k$. For $k = 1, \cdots, L$, $f_k^\theta(v)$ is computed as

$$f_k^\theta(v) = \begin{cases} \sigma\left(W_1^\top v + b_1\right) & k = 1 \\ \sigma\left(W_k^\top f_{k-1}(v) + b_k\right) & k = 2, 3, ..., L - 1 \\ \psi\left(W_L^\top f_{L-1}(v) + b_L\right) & k = L. \end{cases} \tag{4}$$

The policy $\pi_\theta \in \mathbb{R}^{|\mathcal{S}| \times |\mathcal{A}|}$ parametrized by $\theta$ is the output of the final layer:

$$\pi_\theta(\cdot \mid s) = f_L^\theta(\phi(s)) \in \Delta_\mathcal{A}, \quad \forall s \in \mathcal{S}.$$

Our analysis relies two assumptions about the structure of the neural network. The first concerns the invertibility of $\sigma(\cdot)$ as well as the continuity and uniqueness of its inverse, which can be guaranteed by the following:

**Assumption 2** $\sigma$ is strictly monotonic and $\sigma(\mathbb{R}) = \mathbb{R}$. In addition, there do not exist non-zero scalars $\{p_i, q_i\}_{i=1}^m$ with $q_i \neq q_j$, $\forall i \neq j$ such that for some $m > 0$, $\sigma(x) = \sum_{i=1}^m p_i \sigma(x - q_i)$, $\forall x \in \mathbb{R}$.

We note that this assumption holds for common activation functions including leaky-ReLU and parametric ReLU [Xu et al., 2015].

Our second assumption is that the neural network is sufficiently over-parameterized and that the number of parameters decreases with each layer.

**Assumption 3** The output of the first layer is wider than $2|\mathcal{S}|$, and the width of the network decreases over the layers, i.e.

$$n_1 \geq 2|\mathcal{S}|, \text{ and } n_1 > n_2 > ... > n_L = |\mathcal{A}|.$$

Neural networks meeting this criteria have a number of weight parameters that is larger than the cardinality of the state space, making them impractical for large $|\mathcal{S}|$. While ongoing work seeks to relax or remove this assumption, we point out that similar over-parameterization assumptions are critical and very common in most existing works on the theory of neural networks [Zou and Gu, 2019, Nguyen, 2019, Liu et al., 2022, Martinetz and Martinetz, 2022, Pandey and Kumar, 2023].

The $\lambda$-superlevel set of the value function with respect to $\theta$ under reward function $r$ is

$$\mathcal{U}_{\lambda,r}^\Omega \triangleq \{\theta \in \Omega \mid J_r(\pi_\theta) \geq \lambda\}.$$

Our next main theoretical result guarantees the connectedness of $\mathcal{U}_{\lambda,r}^\Omega$.

**Theorem 2** Under Assumptions 1-3, the superlevel set $\mathcal{U}_{\lambda,r}^\Omega$ is connected for any $\lambda \in \mathbb{R}$. In addition, with $J_{r,\Omega}(\theta) \triangleq J_r(\pi_\theta)$, the collection of functions $\{J_{r,\Omega}(\cdot) : \Omega \to \mathbb{R}\}_{r \in \mathbb{R}^{|\mathcal{S}| \times |\mathcal{A}|}}$ is equiconnected.

The proof of this theorem is deferred to the appendix. Similar to Theorem 1, the claim in Theorem 2 on the equiconnectedness of $\{J_{r,\Omega}\}_{r \in \mathbb{R}^{|\mathcal{S}| \times |\mathcal{A}|}}$ is again stronger than the connectedness of $\mathcal{U}_{\lambda,r}^\Omega$ and needs to be derived for the application to minimax theorems, which we discuss in the next section.

## 4 Application to Robust Reinforcement Learning

In this section, we consider the robust RL problem under adversarial reward attack, which can be formulated as a convex-nonconcave minimax optimization program. In Section 4.1, we show that the minimax equality holds for this optimization program in the tabular policy setting and under policies represented by a class of neural networks, as a consequence of our results in Sections 2 and 3. To our best knowledge, the existence of the Nash equilibrium for this robust RL problem has not been established before even in the tabular case. A specific example of this type of robust RL problems is given in Section 4.2.

### 4.1 Minimax Theorem

Robust RL in general studies identifying a policy with reliable performance under uncertainty or attacks. A wide range of formulations have been proposed for robust RL (which we reviewed in details in Section 1.2), and an important class of formulations takes the form of defending against an adversary that can modify the reward function in a convex manner. Specifically, the objective of the learning agent can be described as solving the following minimax optimization problem

$$\max_{\pi \in \Delta_{\mathcal{A}}^{\mathcal{S}}} \min_{r \in \mathcal{C}} J_r(\pi), \tag{5}$$

where $\mathcal{C}$ is some convex set. It is unclear from the existing literature whether minimax inequality holds for this problem, i.e.

$$\max_{\pi \in \Delta_{\mathcal{A}}^{\mathcal{S}}} \min_{r \in \mathcal{C}} J_r(\pi) = \min_{r \in \mathcal{C}} \max_{\pi \in \Delta_{\mathcal{A}}^{\mathcal{S}}} J_r(\pi), \tag{6}$$

and we provide a definitive answer to this question. We note that there exists a classic minimax theorem on a special class of convex-nonconcave functions [Simons, 1995], which we adapt and simplify as follows.

**Theorem 3** *Consider a separately continuous function $f : \mathcal{X} \times \mathcal{Y} \to \mathbb{R}$, with $\mathcal{Y}$ being a convex, compact set. Suppose that $f(x, \cdot)$ is convex for all $x \in \mathcal{X}$. Also suppose that the collection of functions $\{f(\cdot, y)\}_{y \in \mathcal{Y}}$ is equiconnected. Then, we have*

$$\sup_{x \in \mathcal{X}} \min_{y \in \mathcal{Y}} f(x, y) = \min_{y \in \mathcal{Y}} \sup_{x \in \mathcal{X}} f(x, y). \tag{7}$$

Theorem 3 states that the minimax equality holds under two main conditions (besides the continuity condition, which can easily be verified to hold for $J_r(\pi)$). First, the function $f(x, y)$ needs to be convex with respect to the variable $y$ within a convex, compact constraint set. Second, $f(x, y)$ needs to have a connected superlevel set with respect to $x$, and the path function constructed to prove the connectedness of the superlevel set is independent of $y$. As we have shown in this section and earlier in the paper, if we model $J_r(\pi)$ by $f(x, y)$ with $\pi$ and $r$ corresponding to $x$ and $y$, both conditions are observed by the optimization problem (5), which allows us to state the following corollary.

**Corollary 1** *Suppose that the Markov chain $\mathcal{M}$ satisfies Assumption 1 on ergodicity. Then, the minimax equality (6) holds.*

When the neural network presented in Section 3 is used to represent the policy, the collection of functions $\{J_{r,\Omega}\}_r$ is also equiconnected. This allows us to extend the minimax equality above to the neural network policy class. Specifically, consider problem (5) where the policy $\pi_\theta$ is represented by the parameter $\theta \in \Omega$ as described in Section 3. Using $f(x, y)$ to model $J_r(\pi_\theta)$ with $x$ and $y$ mirroring $\theta$ and $r$, we can easily establish the minimax theorem in this case as a consequence of Theorem 2 and 3.

**Corollary 2** *Suppose that the Markov chain $\mathcal{M}$ satisfies Assumption 1 on ergodicity and that the neural policy class satisfies Assumptions 2-3. Then, we have*

$$\sup_{\theta \in \Omega} \min_{r \in \mathcal{C}} J_r(\pi_\theta) = \min_{r \in \mathcal{C}} \sup_{\theta \in \Omega} J_r(\pi_\theta). \tag{8}$$

Corollary 1 and 2 establish the minimax equality (or equivalently, the existence of the Nash equilibrium) for the robust reinforcement learning problem under adversarial reward attack for the tabular and neural network policy class, respectively. To our best knowledge, these results are both novel and previously unknown in the existing literature. The Nash equilibrium is an important global optimality notion in minimax optimization, and the knowledge on its existence can provide strong guidance on designing and analyzing algorithms for solving the problem.

### 4.2 Example - Defense Against Reward Poisoning

We now discuss a particular example of (5). We consider the infinite horizon, average reward MDP $\mathcal{M} = (\mathcal{S}, \mathcal{A}, \mathcal{P}, r)$ introduced in Section 2, where $r$ is the true, unpoisoned reward function. Let $\Pi^{\text{det}}$ denote the set of deterministic policies from $\mathcal{S}$ to $\mathcal{A}$. With the perfect knowledge of this MDP, an attacker has a target policy $\pi_\dagger \in \Pi^{\text{det}}$ and tries to make the learning agent adopt the policy by

manipulating the reward function. Mathematically, the goal of the attacker can be described by the function $\text{Attack}(r, \pi_\dagger, \epsilon_\dagger)$ which returns a poisoned reward under the true reward $r$, the target policy $\pi_\dagger$, and a pre-selected margin parameter $\epsilon_\dagger \geq 0$. $\text{Attack}(r, \pi_\dagger, \epsilon_\dagger)$ is the solution to the following optimization problem

$$
\begin{aligned}
\text{Attack}(r, \pi_\dagger, \epsilon_\dagger) \quad = \quad &\underset{r'}{\arg\min} \quad \sum_{s \in \mathcal{S}, a \in \mathcal{A}} \left( r'(s,a) - r(s,a) \right)^2 \\
&\text{s.t.} \quad J_{r'}(\pi_\dagger) \geq J_{r'}(\pi) + \epsilon_\dagger, \quad \forall \pi \in \Pi^{\text{det}} \backslash \pi_\dagger.
\end{aligned}
\tag{9}
$$

In other words, the attacker needs to minimally modify the reward function to make $\pi_\dagger$ the optimal policy under the poisoned reward. This optimization program minimizes a quadratic loss under a finite number of linear constraints and is obviously convex.

The learning agent observes the poisoned reward $r_\dagger = \text{Attack}(r, \pi_\dagger, \epsilon_\dagger)$ rather than the original reward $r$. As noted in Banihashem et al. [2021], without any defense, the learning agent solves the policy optimization problem under $r_\dagger$ to find $\pi_\dagger$, which may perform arbitrarily badly under the original reward. One way to defend against the attack is to maximize the performance of the agent in the worst possible case of the original reward, which leads to solving a minimax optimization program of the form

$$
\max_{\pi \in \Delta_{\mathcal{A}}^{\mathcal{S}}} \min_{r'} J_{r'}(\pi) \quad \text{s.t. } \text{Attack}(r', \pi_\dagger, \epsilon_\dagger) = r_\dagger.
\tag{10}
$$

When the policy $\pi$ is fixed, (10) reduces to

$$
\min_{r'} J_{r'}(\pi) \quad \text{s.t. } \text{Attack}(r', \pi_\dagger, \epsilon_\dagger) = r_\dagger.
\tag{11}
$$

With the justification deferred to Appendix D, we point out that (11) consists of a linear objective function and a convex (and compact) constraint set, and is therefore a convex program. On the other side, when we fix the reward $r'$, (10) reduces to a standard policy optimization problem.

We are interested in investigating whether the following minimax equality holds.

$$
\max_{\pi \in \Delta_{\mathcal{A}}^{\mathcal{S}}} \min_{r':\text{Attack}(r', \pi_\dagger, \epsilon_\dagger)=r_\dagger} J_{r'}(\pi) = \min_{r':\text{Attack}(r', \pi_\dagger, \epsilon_\dagger)=r_\dagger} \max_{\pi \in \Delta_{\mathcal{A}}^{\mathcal{S}}} J_{r'}(\pi).
\tag{12}
$$

This is a special case of (5) with $\mathcal{C} = \{r' \mid \text{Attack}(r', \pi_\dagger, \epsilon_\dagger) = r_\dagger\}$, which can be verified to be a convex set. Therefore, the validity of (12) directly follows from Corollary 1. Similarly, in the setting of neural network parameterized policy we can establish

$$
\max_{\theta \in \Omega} \min_{r':\text{Attack}(r', \pi_\dagger, \epsilon_\dagger)=r_\dagger} J_{r'}(\pi_\theta) = \min_{r':\text{Attack}(r', \pi_\dagger, \epsilon_\dagger)=r_\dagger} \max_{\theta \in \Omega} J_{r'}(\pi_\theta)
$$

as a result of Corollary 2.

## 5 Conclusions & Future Work

We study the superlevel set of the policy optimization problem under the MDP framework and show that it is always a connected set under a tabular policy and for policies parameterized by a class of neural networks. We apply this result to derive a previously unknown minimax theorem for a robust RL problem. An immediate future direction of the work is to investigate whether/how the result discussed in this paper can be used to design better RL algorithms. In Fatkhullin and Polyak [2021], the authors show that the original LQR problem has connected level sets, but the partially observable LQR does not. It is interesting to study whether this observation extends to the MDP setting, i.e. the policy optimization problem under a partially observable MDP can be shown to have disconnected superlevel sets.

## Acknowledgement

This work was supported in part by the NSF AI Institute AI4OPT, NSF 2112533.

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

# A Proof of Theorems

## A.1 Proof of Theorem 1:

We note that there exists a bijective map between $\pi$ and $\widehat{\mu}_\pi$ where $\widehat{\mu}_\pi$ is induced by $\pi$ according to (2) and conversely

$$\pi(a \mid s) = \frac{\widehat{\mu}_\pi(s,a)}{\mu_\pi(s)} = \frac{\widehat{\mu}_\pi(s,a)}{\sum_{a \in \mathcal{A}} \widehat{\mu}_\pi(s,a)}, \tag{13}$$

provided that $\mu_\pi(s) \neq 0$, which is guaranteed by Assumption 1. Eq. (13) inspires the construction of the path map.

To prove that the superlevel set is connected, we show that for any $\lambda \in \mathbb{R}$ and $\pi_1, \pi_2 \in \mathcal{U}_{\lambda,r}$, there exists a continuous path map $p : [0,1] \to \mathcal{U}_{\lambda,r}$ such that $p(0) = \pi_1$ and $p(1) = \pi_2$. We now construct the path function $p$ by defining

$$p(\alpha)(a \mid s) = \frac{\alpha \mu_{\pi_1}(s)\pi_1(a \mid s) + (1-\alpha)\mu_{\pi_2}(s)\pi_2(a \mid s)}{\alpha \mu_{\pi_1}(s) + (1-\alpha)\mu_{\pi_2}(s)},$$

which is well-defined for all $\alpha \in [0,1]$ as $\mu_{\pi_1}(s), \mu_{\pi_2}(s)$ are positive for all $s \in \mathcal{S}$. Note that the construction of $p$ does not depend on the reward function $r$. It is easy to see that $p(\alpha) \in \Delta_{\mathcal{A}}^{\mathcal{S}}$ is a continuous in $\alpha$. To stress that $p(\alpha)$ is in the policy space, we denote $\pi_\alpha = p(\alpha)$.

Recall the definition of the transition probability matrix in (1). We define $B \in \mathbb{R}^{|\mathcal{S}|}$ as

$$B = P^{\pi_\alpha} \cdot (\alpha \mu_{\pi_1} + (1-\alpha)\mu_{\pi_2}).$$

Each entry of $B$ can be expressed as

$$B(s') = \sum_{s,a} \mathcal{P}(s' \mid s,a)\pi_\alpha(a \mid s)(\alpha \mu_{\pi_1}(s) + (1-\alpha)\mu_{\pi_2}(s))$$

$$= \sum_{s,a} \mathcal{P}(s' \mid s,a)\frac{\alpha \mu_{\pi_1}(s)\pi_1(a \mid s) + (1-\alpha)\mu_{\pi_2}(s)\pi_2(a \mid s)}{\alpha \mu_{\pi_1}(s) + (1-\alpha)\mu_{\pi_2}(s)}(\alpha \mu_{\pi_1}(s) + (1-\alpha)\mu_{\pi_2}(s))$$

$$= \sum_{s,a} \mathcal{P}(s' \mid s,a)\alpha \mu_{\pi_1}(s)\pi_1(a \mid s) + \sum_{s,a} \mathcal{P}(s' \mid s,a)(1-\alpha)\mu_{\pi_2}(s)\pi_2(a \mid s)$$

$$= \alpha \sum_{s,a} P^{\pi_1}_{s',s}\mu_{\pi_1}(s) + (1-\alpha)\sum_{s,a} P^{\pi_2}_{s',s}\mu_{\pi_2}(s)$$

$$= \alpha \mu_{\pi_1}(s') + (1-\alpha)\mu_{\pi_2}(s'),$$

which implies

$$P^{\pi_\alpha} \cdot (\alpha \mu_{\pi_1} + (1-\alpha)\mu_{\pi_2}) = \alpha \mu_{\pi_1} + (1-\alpha)\mu_{\pi_2}. \tag{14}$$

A consequence of Assumption 1 is that for any policy $\pi$ there is a unique eigenvector of $P^\pi$ associated with the eigenvalue 1, and this eigenvector (properly normalized) is the stationary distribution. Therefore, (14) means that $\alpha \mu_{\pi_1} + (1-\alpha)\mu_{\pi_2}$ has to be the stationary distribution under policy $\pi_\alpha$, i.e.

$$\mu_{\pi_\alpha} = \alpha \mu_{\pi_1} + (1-\alpha)\mu_{\pi_2}.$$

As a result, for all $s \in \mathcal{S}, a \in \mathcal{A}$

$$\widehat{\mu}_{\pi_\alpha}(s,a) = \mu_{\pi_\alpha}(s)\pi_\alpha(a \mid s)$$

$$= (\alpha \mu_{\pi_1}(s) + (1-\alpha)\mu_{\pi_2}(s))\frac{\alpha \mu_{\pi_1}(s)\pi_1(a \mid s) + (1-\alpha)\mu_{\pi_2}(s)\pi_2(a \mid s)}{\alpha \mu_{\pi_1}(s) + (1-\alpha)\mu_{\pi_2}(s)}$$

$$= \alpha \mu_{\pi_1}(s)\pi_1(a \mid s) + (1-\alpha)\mu_{\pi_2}(s)\pi_2(a \mid s)$$

$$= \alpha \widehat{\mu}_{\pi_1}(s,a) + (1-\alpha)\widehat{\mu}_{\pi_2}(s,a).$$

Note that $J_r(\pi) = \sum_{s \in \mathcal{S}, a \in \mathcal{A}} r(s,a)\widehat{\mu}_\pi(s,a)$. Since $\pi_{\pi_1}, \pi_{\pi_2} \in \mathcal{U}_{\lambda,r}$, we know

$$\sum_{s \in \mathcal{S}, a \in \mathcal{A}} r(s,a)\widehat{\mu}_{\pi_1}(s,a) \geq \lambda, \qquad \sum_{s \in \mathcal{S}, a \in \mathcal{A}} r(s,a)\widehat{\mu}_{\pi_2}(s,a) \geq \lambda.$$

Therefore, we have for any $\alpha \in [0, 1]$

$$J_r(\pi_\alpha) = \sum_{s \in \mathcal{S}, a \in \mathcal{A}} r(s, a)\widehat{\mu}_{\pi_\alpha}(s, a) = \sum_{s \in \mathcal{S}, a \in \mathcal{A}} r(s, a)\left(\alpha\widehat{\mu}_{\pi_1}(s, a) + (1 - \alpha)\widehat{\mu}_{\pi_2}(s, a)\right) \geq \lambda,$$

which implies $\pi_\alpha \in \mathcal{U}_{\lambda,r}$. So far we have verifed that the constructed path map $p$ is indeed continuous and maps $\alpha \in [0, 1]$ to $\mathcal{U}_{\lambda,r}$ with $p(0) = \pi_1$ and $p(1) = \pi_2$. This concludes the proof on the connectedness of the superlevel set $\mathcal{U}_{\lambda,r}$. The claim on the equiconnectedness simply follows from the fact that the construction of the path map $p$ does not depend on the reward function.

∎

## A.2   Proof of Theorem 2

We use $X$ to denote the concatenation of the feature vectors across all states

$$X \triangleq \begin{bmatrix} \phi(s_1)^\top \\ \phi(s_2)^\top \\ \vdots \\ \phi(s_{|\mathcal{S}|})^\top \end{bmatrix} \in \mathbb{R}^{|\mathcal{S}| \times d}$$

In the analysis we may apply the softmax function $\psi$ to a matrix in a row-wise fashion. Specifically, for any $n \geq 1$ and matrix $M \in \mathbb{R}^{n \times |\mathcal{A}|}$, we have

$$\psi(M)_{i,j} = \frac{\exp(M_{i,j})}{\sum_{j'=1}^{|\mathcal{A}|} \exp(M_{i,j'})} \quad \forall i = 1, ..., n.$$

The softmax operator $\psi$ can be inverted up to an additive constant factor. We define $\psi_{inv}$ for any matrix $M \in \mathbb{R}^{n \times |\mathcal{A}|}$ as

$$\psi_{inv}(M)_{i,j} = \log(M_{i,j}) + c_i \quad \forall i, j,$$

with $c_i$ determined such that $\sum_{j=1}^{|\mathcal{A}|} \psi_{inv}(M)_{i,j} = 0$. Note that $\psi_{inv}$ is a right inverse of $\psi$, i.e. $\psi(\psi_{inv}(M)) = M$ for all matrix $A$.

When the input to a neural network with parameter $\theta$ is the feature table $X$, we denote the output of layer $k$ by $F_k^\theta \in \mathbb{R}^{|\mathcal{S}| \times n_k}$. According to (4), $F_k^\theta$ can be expressed as

$$F_k^\theta = \begin{cases} \sigma\left(XW_1 + \mathbf{1}_{|\mathcal{S}|}b_1^\top\right) & k = 1 \\ \sigma\left(F_{k-1}^\theta W_k + \mathbf{1}_{|\mathcal{S}|}b_k^\top\right) & k = 2, 3, ..., L - 1 \\ \psi\left(F_{L-1}^\theta W_L + \mathbf{1}_{|\mathcal{S}|}b_L^\top\right) & k = L \end{cases}$$

where $\mathbf{1}_{|\mathcal{S}|}$ is the all-one vector of dimension $|\mathcal{S}| \times 1$. Note that $F_L^\theta \in \mathbb{R}^{|\mathcal{S}| \times |\mathcal{A}|}$ is the policy table produced by the neural network, i.e. $\pi_\theta = F_L^\theta$.

The proof of Theorem 2 relies on the following intermediate results, which we now present. The proof of Proposition 1 can be found in Appendix B.

**Proposition 1** *If* $\mathrm{rank}(X) = |\mathcal{S}|$*, then under Assumption 1 and 2, the superlevel set* $\mathcal{U}_{\lambda,r}^\Omega$ *is connected for all* $\lambda \in \mathbb{R}$*.*

**Lemma 1** *Let* $(X, W, b, V) \in \mathbb{R}^{|\mathcal{S}| \times n_0} \times \mathbb{R}^{n_0 \times n_1} \times \mathbb{R}^{n_1} \times \mathbb{R}^{n_1 \times n_2}$*. Let* $Z = \sigma(XW + \mathbf{1}_{|\mathcal{S}|}b^\top)V$*. Suppose* $X$ *has distinct rows. Then, under Assumption 2 and 3, there exists a continuous path map* $c : [0, 1] \to \mathbb{R}^{n_0 \times n_1} \times \mathbb{R}^{n_1} \times \mathbb{R}^{n_1 \times n_2}$ *with* $c(\lambda) = (W(\lambda), b(\lambda), V(\lambda))$ *such that*

*1)* $c(0) = (W,b,V)$,

*2)* $\sigma\left(XW(\lambda) + \mathbf{1}_{|\mathcal{S}|}b(\lambda)^T\right)V(\lambda) = Z, \forall \lambda \in [0, 1]$,

*3)* $\mathrm{rank}\left(\sigma\left(XW(1) + \mathbf{1}_{|\mathcal{S}|}b(1)^T\right)\right) = N$.

**Lemma 2** *Let* $(X, W, V, W') \in \mathbb{R}^{|\mathcal{S}| \times n_0} \times \mathbb{R}^{n_0 \times n_1} \times \mathbb{R}^{n_1 \times n_2} \times \mathbb{R}^{n_0 \times n_1}$*. Suppose* $\mathrm{rank}(\sigma(XW)) = |\mathcal{S}|$ *and* $\mathrm{rank}(\sigma(XW')) = |\mathcal{S}|$*. Then, under Assumption 2 and 3, there exists a continuous path map* $c : [0, 1] \to \mathbb{R}^{n_0 \times n_1} \times \mathbb{R}^{n_1 \times n_2}$ *with* $c(\lambda) = (W(\lambda), V(\lambda))$ *such that*

*1) c(0) = (W,V),*

*2) $\sigma(XW(\lambda))V(\lambda) = \sigma(XW)V, \forall \lambda \in [0,1]$,*

*3) $W(1) = W'$.*

To prove Theorem 2, it suffices to show that for any $\theta_1 = (W_{1,l}, b_{1,l})_{l=1}^L \in \mathcal{U}_{\lambda,r}^\Omega$ and $\theta_2 = (W_{2,l}, b_{2,l})_{l=1}^L \in \mathcal{U}_{\lambda,r}^\Omega$ there exists a connected path that is completely within $\mathcal{U}_{\lambda,r}^\Omega$.

Applying Lemma 1 with $(X, W_{1,1}, b_{1,1}, W_{1,2})$ and $(X, W_{2,1}, b_{2,1}, W_{2,2})$, the problem simplifies to showing the existence of a continuous path within $\mathcal{U}_{\lambda,r}^\Omega$ that connects

$$\theta_1' = ((W_{1,1}', b_{1,1}'), (W_{1,2}', b_{1,2}), (W_{1,l}, b_{1,l})_{l=3}^L)$$

and

$$\theta_2' = ((W_{2,1}', b_{2,1}'), (W_{2,2}', b_{1,2}), (W_{2,l}, b_{2,l})_{l=3}^L)$$

such that

$$\text{rank}(F_1^{\theta_1'}) = \text{rank}(F_1^{\theta_2'}) = |\mathcal{S}|.$$

Then, we can apply Lemma 2 with $([X, 1_{|\mathcal{S}|}], [W_{1,1}'^\top, b_{1,1}']^\top, W_{1,2}', [W_{2,1}'^\top, b_{2,1}']^\top)$ to show that there is a continuous path between $\theta_1'$ and $\theta_1''$ with $\theta_1'' = ((W_{2,1}', b_{2,1}'), (W_{1,2}'', b_{1,2}), (W_{1,l}, b_{1,l})_{l=3}^L)$ such that

$$\text{rank}(F_1^{\theta_1''}) = \text{rank}(F_1^{\theta_1'}) = |\mathcal{S}|.$$

As a consequence, now we simply have to show that $\theta_1''$ and $\theta_2'$ is connected by a continuous path within $\mathcal{U}_{\lambda,r}^\Omega$.

Note that $\theta_1''$ and $\theta_2'$ have identical first layer parameters and thus the same first layer output, which is full rank. This allows us to treat the layers from 2 to $L$ as a new network and apply Proposition 1 (which requires the input to be full rank) to the new network to guarantee that there exists a continuous path map $c : [0,1] \to \Omega_2 \times ... \times \Omega_k$ such that $c(0) = ((W_{1,2}'', b_{1,2}), (W_{1,l}, b_{1,l})_{l=3}^L)$, $c(1) = ((W_{2,2}', b_{1,2}), (W_{2,l}, b_{2,l})_{l=3}^L)$, and

$$\min\{J_r(\pi_{\theta_1}), J_r(\pi_{\theta_2})\} \leq J_r(\pi_{((W_{2,1}', b_{2,1}'), c(\alpha))}) \leq \max\{J_r(\pi_{\theta_1}), J_r(\pi_{\theta_2})\}$$

for all $\alpha \in [0,1]$. This implies that there is indeed a continuous path between $\theta_1''$ and $\theta_2'$ within $\mathcal{U}_{\lambda,r}^\Omega$.

Similar to the proof of Theorem 1, the claim on the connectedness simply follows from the fact that the construction of the path map $p$ does not depend on the reward function. ∎

# B   Proof of Proposition 1

For each layer of the neural network $k = 1, \cdots, L$, we define $\Omega_k^\star \subseteq \Omega_k$ to be the set of weights $W_k$ and biases $b_k$ of layer $k$ such that $W_k$ is full rank, i.e.

$$\Omega_k^\star = \{(W_k, b_k) \in \Omega_k : W_k \text{ is full rank}\}. \tag{15}$$

We denote $\Omega^\star = \Omega_1^\star \times \Omega_2^\star \times ... \times \Omega_L^\star$. Next, we introduce the following lemmas in aid of the analysis.

**Condition 1** *Given $\theta = (W_l, b_l)_{l=2}^L$, $W_l$ has full rank for every $l \in [2, L]$.*

**Lemma 3** *Under Assumption 2, 3, and Condition 1, given any $k \in [2, L]$ and matrix $F \in \mathbb{R}^{|\mathcal{S}| \times n_k}$, there exists a continuous map $h : \Omega_2^\star \times ... \times \Omega_k^\star \times \mathbb{R}^{|\mathcal{S}| \times n_k} \to \Omega_1$ such that*

*1) Given $((W_2, b_2), ..., (W_k, b_k), F) \in \Omega_2^\star \times ... \times \Omega_k^\star \times \mathbb{R}^{|\mathcal{S}| \times n_k}$, we have*

$$F_k^{h((W_l, b_l)_{l=2}^k, F), (W_l, b_l)_{l=2}^k} = F.$$

*2) For any $\theta^\star = (W_l^\star, b_l^\star)_{l=1}^L \in \Omega_1 \times \Omega_2^\star \times ... \times \Omega_L^\star$, there exists a continuous path map $p : [0,1] \to \Omega_1 \times \Omega_2^\star \times ... \times \Omega_L^\star$ such that $p(0) = \theta^\star$, $p(1) = \left(h((W_l^\star, b_l^\star)_{l=2}^k, F_k^{\theta^\star}), (W_l^\star, b_l^\star)_{l=2}^L\right)$, and $F_L^{p(\alpha)} = F_L^{\theta^\star}$ for all $\alpha \in [0,1]$.*

**Lemma 4** *Given two connected sets $\mathcal{A} \subseteq \mathbb{R}^{m_1 \times n}$ and $\mathcal{B} \subseteq \mathbb{R}^{n \times m_2}$, the set $\{ab \mid a \in \mathcal{A}, b \in \mathcal{B}\}$ is connected. Given two connected sets $\mathcal{A}, \mathcal{B} \subseteq \mathbb{R}^{m \times n}$, the set $\{a + b \mid a \in \mathcal{A}, b \in \mathcal{B}\}$ is connected.*

**Lemma 5** *Under Assumption 2, for any $\theta \in \Omega$, there exist $\theta^\star \in \Omega^\star$ and a continuous path map $p : [0, 1] \to \Omega$ such that $p(0) = \theta$, $p(1) = \theta^\star$, and $F_L^{p(\alpha)} = F_L^\theta$ for all $\alpha \in [0, 1]$.*

**Lemma 6** *If $n < m$, then the set $\mathcal{F} = \{F \in \mathbb{R}^{m \times n} \mid \mathrm{rank}(F) = n\}$ is connected. In other words, given $F_1, F_2 \in \mathcal{F}$, there exists a continuous path map $q : [0, 1] \to \mathcal{F}$ such that $q(0) = F_1$ and $q(1) = F_2$.*

Fix a $\lambda \in \mathbb{R}$. To show the superlevel set $\mathcal{U}_{\lambda,r}^\Omega$ is connected, it suffices to show that for any $\theta_1, \theta_2 \in \mathcal{U}_{\lambda,r}^\Omega$, there exists a continuous path between them that is completely in $\mathcal{U}_{\lambda,r}^\Omega$.

Without any loss of generality, we can safely assume that both $\theta_1 = (W_{1,l}, b_{1,l})_{l=1}^L$ and $\theta_2 = (W_{2,l}, b_{2,l})_{l=1}^L$ satisfy Condition 1, since otherwise by Lemma 5 we can find a continuous path from $\theta_1$ and $\theta_2$ that leads to one satisfying Condition 1. We denote the policies parameterized by $\theta_1, \theta_2$ as $\pi_1, \pi_2$, i.e.

$$\pi_1 = F_L^{\theta_1}, \quad \pi_2 = F_L^{\theta_2}.$$

By Lemma 3, there is a continuous path from $\theta_1/\theta_2$ to $\theta_1'/\theta_2'$ where we define

$$\theta_1' = \left( h\left( (W_{1,l}, b_{1,l})_{l=2}^L, \pi_1 \right), (W_{1,l}, b_{1,l})_{l=2}^L \right),$$
$$\text{and} \quad \theta_2' = \left( h\left( (W_{2,l}, b_{2,l})_{l=2}^L, \pi_2 \right), (W_{2,l}, b_{2,l})_{l=2}^L \right).$$

Now, we just have to show that there exists a continuous path between $\theta_1'$ and $\theta_2'$ that is completely within $\mathcal{U}_{\lambda,r}^\Omega$. By Lemma 6, we know that for $l = 2, .., L$, there exists a continuous path map $q_l : [0, 1] \to \Omega_l^\star$ such that $q_l(1) = W_{1,l}$ and $q_l(0) = W_{2,l}$. Then, we construct the map $q : [0, 1] \to \Omega$

$$q(\alpha) = \left( h((q_l(\alpha), \alpha b_{1,l} + (1 - \alpha)b_{2,l})_{l=2}^L, \pi_1), (q_l(\alpha), \alpha b_{1,l} + (1 - \alpha)b_{2,l})_{l=2}^L \right) \quad \forall \alpha \in [0, 1].$$

It is obvious that $q$ is a continuous map as $h, q_2, ..., q_L$ are continuous. In addition, $F_L^{q(\alpha)} = \pi_1$ for all $\alpha \in [0, 1]$, and $q(1) = \theta_1'$. We define

$$\theta_1'' = q(0) = \left( h\left( (W_{2,l}, b_{2,l})_{l=2}^L, \pi_1 \right), (W_{2,l}, b_{2,l})_{l=2}^L \right).$$

Now our aim simplifies to finding a continuous path between $\theta_1''$ and $\theta_2'$ that is completely in $\mathcal{U}_{\lambda,r}^\Omega$. To show that this path exists, we construct a continuous map $t : [0, 1] \to \Omega$ as follows

$$t(\alpha) = \left( h\left( (W_{2,l}, b_{2,l})_{l=2}^L, \tilde{\pi}_\alpha \right), (W_{2,l}, b_{2,l})_{l=2}^L \right) \quad \forall \alpha \in [0, 1],$$

where $\tilde{\pi}$ is defined entry-wise

$$\tilde{\pi}_\alpha(a \mid s) = \frac{\alpha \mu_{\pi_1}(s)\pi_1(a \mid s) + (1 - \alpha)\mu_{\pi_2}(s)\pi_2(a \mid s)}{\alpha \mu_{\pi_1}(s) + (1 - \alpha)\mu_{\pi_2}(s)}.$$

It can be seen that $t$ is indeed continuous since $\tilde{\pi}_\alpha$ is continuous in $\alpha$, and $t(0) = \theta_2'$ and $t(1) = \theta_1''$. What remains to be shown is that $F_L^{t(\alpha)} \in \mathcal{U}_{\lambda,r}^\Omega$, i.e. $J_r(F_L^{t(\alpha)}) \geq \lambda$. By the definition of $h$ in Lemma 3, $F_L^{t(\alpha)} = \tilde{\pi}_\alpha$. It has been shown in the proof of Theorem 1 that indeed $J_r(\tilde{\pi}_\alpha) \geq \lambda$ provided that $J_r(\pi_1) \geq \lambda$ and $J_r(\pi_2) \geq \lambda$. This concludes the proof of Proposition 1.

$\blacksquare$

# C  Proof of Supporting Lemmas

## C.1  Proof of Lemma 1

This lemma is adapted from Lemma 5.2 of Nguyen [2019].

## C.2 Proof of Lemma 2

This lemma is adapted from Lemma 5.3 of Nguyen [2019].

## C.3 Proof of Lemma 3

We provide a proof for the case $k = L$. For $k \neq L$, the proof can be found in Nguyen [2019][Lemma 3.3].

For $((W_2, b_2), ..., (W_L, b_L), \pi) \in \Omega_2^\star \times ... \times \Omega_L^\star \times \Delta_{\mathcal{A}}^{\mathcal{S}}$, we define the map $h$ as follows

$$h\left((W_l, b_l)_{l=2}^L, \pi\right) = \left(\widehat{W}_1, \widehat{b}_1\right)$$

where $\widehat{W}_1$ and $\widehat{b}_1$ is defined as

$$\begin{cases} \begin{bmatrix} W_1 \\ b_1^\top \end{bmatrix} = [X, \mathbf{1}_{|\mathcal{S}|}]^\dagger \sigma^{-1}(B_1), \\ B_l = \left(\sigma^{-1}(B_{l+1}) - \mathbf{1}_{|\mathcal{S}|} b_{l+1}^\top\right) W_{l+1}^\dagger, \forall l \in [1, k-2] \\ B_{k-1} = \left(\psi_{inv}(\pi) - \mathbf{1}_{|\mathcal{S}|} b_L^\top\right) W_L^\dagger \end{cases} \tag{16}$$

where we use $A^\dagger$ to denote the Moore-Penrose inverse of a matrix $A$. If A has full column rank, then we have $A^\dagger A = I$. If A has full row rank, we have $AA^\dagger = I$. We can easily see that the defined $h$ operator is continuous as it is a composition of continuous operators.

Assumption 3, and Condition 1 imply that the matrices $W_2, ..., W_L$ all have full column rank, which means $W_l^\dagger W_l = I$. We also know that $[X, \mathbf{1}_{|\mathcal{S}|}]$ has full row rank by our assumption that $X$ has full row rank, which means $[X, \mathbf{1}_{|\mathcal{S}|}][X, \mathbf{1}_{|\mathcal{S}|}]^\dagger = I$. Therefore, we can layerwise invert (16) and verify that

$$F_L^{h((W_l, b_l)_{l=2}^L, \pi), (W_l, b_l)_{l=2}^L} = \pi.$$

For every layer $l = 2, ..., L$, we define the operator $G_l : \mathbb{R}^{|\mathcal{S}| \times n_{l-1}} \to \mathbb{R}^{|\mathcal{S}| \times n_l}$

$$G_l(Y) = \begin{cases} \sigma\left(Y W_l^\star + \mathbf{1}_{|\mathcal{S}|}(b_l^\star)^\top\right) & l \in [2, L-1] \\ \psi\left(Y W_L^\star + \mathbf{1}_{|\mathcal{S}|}(b_L^\star)^\top\right) & l = L \end{cases}$$

We also define the operator $H : \mathbb{R}^{(d+1) \times n_1} \to \mathbb{R}^{|\mathcal{S}| \times n_1}$

$$H(Y) = \sigma\left([X, \mathbf{1}_{|\mathcal{S}|}] Y\right).$$

To show the continuous path claimed in Lemma 3 exists, it suffices to show that the set $\{(W_1, b_1) : F_L^{(W_1, b_1), (W_l^\star, b_l^\star)_{l=2}^L} = F_L^{\theta^\star}\}$ is connected, which is equivalent to showing that the set $f^{-1}(F_L^{\theta^\star})$ is connected where $f$ is defined as

$$f([W_1^\top, b_1]^\top) = G_L \circ ... \circ G_2 \circ H([W_1^\top, b_1]^\top).$$

Note that the definition of $f$ implies

$$f^{-1}(\pi) = H^{-1} \circ G_2^{-1} \circ ... \circ G_L^{-1}(\pi). \tag{17}$$

Note that $G_l^{-1}$ is

$$G_l^{-1}(F) = \begin{cases} \left(\psi_{inv}(F) + \{C \mid C_{i,j} = C_{i,j'} \,\forall i, j \neq j'\} - \mathbf{1}_N b_L^T\right)(W_L^\star)^\dagger + \{B \mid BW_L^\star = 0\}, & l = L \\ \left(\sigma^{-1}(F) - \mathbf{1}_N b_l^\star\right)(W_l^\star)^\dagger + \{B \mid BW_l^\star = 0\}, & l = 2, ..., L-1 \end{cases}$$

It is easy to verify that $\{C \mid C_{i,j} = C_{i,j'} \,\forall i, j \neq j'\}$ and $\{B \mid BW_l^\star = 0\}$ for all $l = 2, ..., L$. Then, Lemma 4 implies that $G_l^{-1}(F)$ is a connected set for all $F$.

Similarly, $H^{-1}(F) = [X, \mathbf{1}_{|\mathcal{S}|}]^\dagger \sigma^{-1}(F) + \{B \mid [X, \mathbf{1}_{|\mathcal{S}|}]B = 0\}$ is also a connected set for all $F$. Therefore, from (17) we know that $f^{-1}(F)$ is a connected set for any $F$, which concludes the proof of Lemma 3.

■

### C.4 Proof of Lemma 4

To show that the product of the two connected sets are connected, we consider any $x, y \in \{ab \mid a \in \mathcal{A}, b \in \mathcal{B}\}$. Obviously, there exist $a_x, a_y \in \mathcal{A}$ and $b_x, b_y \in \mathcal{B}$ such that $x = a_x b_x$, $y = a_y b_y$. The connectedness of $\mathcal{A}$ and $\mathcal{B}$ implies that there exists continuous path maps $p_\mathcal{A} : [0, 1] \to \mathcal{A}$ and $p_\mathcal{B} : [0, 1] \to \mathcal{B}$ such that $p_\mathcal{A}(0) = a_x$, $p_\mathcal{A}(1) = a_y$, $p_\mathcal{B}(0) = b_x$, $p_\mathcal{B}(1) = b_y$. Define $p(\alpha) = p_\mathcal{A} p_\mathcal{B}$ for $\alpha \in [0, 1]$. It is obvious that $p(\alpha) \in \{ab \mid a \in \mathcal{A}, b \in \mathcal{B}\}$ for all $\alpha$. Since the product of continuous maps is still continuous, $p : [0, 1] \to \{ab \mid a \in \mathcal{A}, b \in \mathcal{B}\}$ is a continuous path map satisfying $p(0) = x$ and $p(1) = y$. This implies that the set $\{ab \mid a \in \mathcal{A}, b \in \mathcal{B}\}$ is a connected set.

A similar argument can be used to show that the sum of two connected sets is connected.

■

### C.5 Proof of Lemma 5

Define $\tilde{F}_L((W_l, b_l)_{l=1}^L)$ as the output of the final layer before the softmax activation

$$\tilde{F}_L^{(W_l, b_l)_{l=1}^L} = F_{L-1} W_L + \mathbf{1}_{|\mathcal{S}|} b_L^\top.$$

Then, existing results in the literature (such as Lemma 3.4 of Nguyen [2019]) show that for any $\theta \in \Omega$, there exists a continuous path map $p : [0, 1] \to \Omega$ such that $p(0) = \theta$, $p(1) = \theta^\star \in \Omega^\star$, and $\tilde{F}_L(p(\alpha)) = \tilde{F}_L(\theta)$ for all $\alpha \in [0, 1]$. This leads to our claim.

■

### C.6 Proof of Lemma 6

This lemma is adapted from Theorem 4 of Evard and Jafari [1994].

## D   Convexity of Optimization Program (11)

In this section, we show that (11) is a convex optimization program. First, we note that

$$J_{r'}(\pi) = \sum_{s,a} r'(s, a) \widehat{\mu}_\pi = \widehat{\mu}_\pi^\top r',$$

which means that the objective function is linear in the reward.

The constraint set is obvious closed. It is also bounded as the reward $r(s, a) \in [0, U_r]$. To prove the constraint set is convex, we need to show that for any $r_1, r_2$ such that $\text{Attack}(r_1, \pi_\dagger, \epsilon_\dagger) = r_\dagger$ and $\text{Attack}(r_2, \pi_\dagger, \epsilon_\dagger) = r_\dagger$, we have

$$\text{Attack}(\alpha r_1 + (1 - \alpha) r_2, \pi_\dagger, \epsilon_\dagger) = r_\dagger, \quad \forall \alpha \in [0, 1]. \tag{18}$$

By the optimality condition of (9), $r_\dagger$ being the optimal poisoned reward for true reward $r_1$ and $r_2$ is equivalent to

$$\langle r - r_\dagger, r_1 - r_\dagger \rangle \leq 0 \quad \text{and} \quad \langle r - r_\dagger, r_2 - r_\dagger \rangle \leq 0$$

for all $r$ such that $J_r(\pi_\dagger) \geq J_r(\pi) + \epsilon_\dagger$, $\forall \pi \in \Pi^{\text{det}} \backslash \pi_\dagger$. By taking the convex combination of these two inequalities, we have for any $\alpha \in [0, 1]$

$$\langle r - r_\dagger, \alpha r_1 + (1 - \alpha) r_2 - r_\dagger \rangle \leq 0 \tag{19}$$

for all $r$ such that $J_r(\pi_\dagger) \geq J_r(\pi) + \epsilon_\dagger$, $\forall \pi \in \Pi^{\text{det}} \backslash \pi_\dagger$. Again by the optimality condition of (9), (19) is equivalent to (18).

At this point, we have shown that (11) has a linear objective function and a convex (and also compact) constraint set. As a result, the optimization program is convex.

