# OpenReview forum: "Connected Superlevel Set in (Deep) Reinforcement Learning and its Application to Minimax Theorems"
_NeurIPS.cc/2023/Conference — NeurIPS 2023 poster_

### Official Review · Reviewer_DdeT · 2023-07-06

**Soundness:** 3 good
**Presentation:** 3 good
**Contribution:** 3 good
**Rating:** 6
**Confidence:** 1

**Summary:**

Authors found novel discoveries in policy optimization problems: 1.the superlevel set of the objective function related to the policy parameter is always a connected set and the optimization objective as a function of the policy parameter and reward satisfies a stronger “equiconnectedness” property. Based on the discoveries, authors derive a novel minimax theorem for a robust RL problem.


**Strengths:**

Well written and well organized.
Novel discoveries of policy optimization problems in reinforcement learning.
Derived a novel minimax theorem for a robust RL problem which may contribute to the development of novel robust algorithm in the future. Theoretical analysis and proofs are nicely presented.


**Weaknesses:**

Checking math is not my strength so I'll refrain from providing my opinion on the math. I think one weakness is about the application side of the paper and lack of any experiments(maybe just a toy example?).

**Questions:**

Would it be possible to design a toy env and run some small experiments to explicitly show how the derived theorem can help develop better algorithm in addition to the reward poisoning example?

**Limitations:**

Again, checking math is not my strength so I'll refrain from providing my opinion on the math. One limitation is lack of experiments and application.

---

> ### Author Rebuttal · Authors · 2023-08-06
>
> We thank the reviewer for the time and effort in reviewing the paper. We agree that it would be good to complement the theory with simulations in some way, but so far the results mostly focus on the fundamental structure of the optimization problem, and it is unclear what simulations would be meaningful. We do want to point out that our theoretical results can lead to the design of better algorithms as we discuss in the response to Reviewer ceyJ.

---

### Official Review · Reviewer_UPhm · 2023-07-07

**Soundness:** 3 good
**Presentation:** 3 good
**Contribution:** 3 good
**Rating:** 7
**Confidence:** 4

**Summary:**

This work shows the superlevel set of the objective function in reinforcement learning is always equiconnected for both tabular policy and neural policy. An application of the connected property is the minimax theorem. As a consequence, reward attack robust RL can be shown to have Nash equilibrium.

**Strengths:**

1. The paper show the superlevel set of the RL objective is connected. Furthermore, the collection of the objective functions under the tabular and the neural policy is equiconnected.
2. The connected property is used to establish the minimax theorem.
3. As a corollary, the authors show that reward attack robust RL problems have Nash equilibrium. This is the first work that shows such a result.
4. The paper is well-written.

**Weaknesses:**

1. Although Theorem 1 and 2 are general results that cover a large class of policies, it is not clear whether the minimax theorem can be used to show the existence of Nash equilibrium for other robust RL problems, e.g., when the agent is uncertain of the transition kernels.
2. It is unclear whether the theoretical results can help design algorithms for robust RL problems.

**Questions:**

I have some minor questions:
1. What is the intuition behind Assumption 2? Why do we need the non-zero scalar condition?
2. Can we use the equiconnected property to prove minimax theorems other than Theorem 3?


**Limitations:**

The authors address limitations in the conclusion section.

---

> ### Author Rebuttal · Authors · 2023-08-06
>
> We sincerely thank the reviewer for the feedback, which we will incorporate when making the next revision. We confirm that our results can be used to drive the design of algorithms (please see our response to Reviewer ceyJ above).
>
> The technical reason for considering Assumption 2 is to make the activation function invertible and its inverse unique, which ensures that the activation function does not break the connected path constructed in the analysis. A sufficient condition that guarantees Assumption 2 is that the activation function is 1) monotonically increasing or decreasing and 2) piecewise linear.
>
> Regarding the question on whether the equiconnectedness property can be used to derive results other than Theorem 3, we would like to point to the pseudo-linearity condition that Reviewer efCX brought up from a recent work [Jin 2022]. In the tabular case, this condition gets implied by our results and analysis of Theorem 1, and it plays an important role in the algorithm design process in [Jin 2022]. Theorem 2 of our paper can be used to establish a neural network version of the pseudo-linearity condition, which may be helpful in extending their algorithm to the function approximation setting.
>
> References
>
> Jin, Yujia, Vidya Muthukumar, and Aaron Sidford. "The complexity of infinite-horizon general-sum stochastic games." arXiv preprint arXiv:2204.04186 (2022).

---

> > ### Comment · Reviewer_UPhm · 2023-08-17
> >
> > Thanks for your response. My questions have been addressed.

---

### Official Review · Reviewer_ceyJ · 2023-07-09

**Soundness:** 3 good
**Presentation:** 4 excellent
**Contribution:** 3 good
**Rating:** 5
**Confidence:** 3

**Summary:**

This paper aims to enhance the comprehension of the optimization landscape in reinforcement learning (RL) for policy optimization problems. The primary contribution of this work is to demonstrate the connectedness of the superlevel set of the policy optimization problem in RL under a tabular policy representation. Furthermore, the authors establish that the superlevel set of the objective function, considering the policy parameters (i.e., weights of the neural networks), remains connected across all levels. The authors also illustrate the practical implications of their main findings by deriving a minimax theorem for a specific class of robust RL problems.

**Strengths:**

- The paper studies an exciting area of research on optimization applied to policy learning in reinforcement learning.
- The paper is generally well written and coherent.

**Weaknesses:**

- The paper briefly talks about robust RL as a potential application of studying super level connectedness. It does not provide any practical robust RL algorithm.

**Questions:**

- I would like the authors to include an explanation for why one needs to expand their understanding of RL optimization landscape beyond gradient domination. To motivate the problem to the general conference audience, it is important to list additional insights one might gain from studying super level set connectedness (SLSC). For example, does SLSC help identifying bottlenecks, regions of poor convergence, or potential areas for algorithmic improvement?
- It is also not clear to me what were the exact challenges in establishing results for deep RL?
- Are the results in Sec 2 similar to Nguyen [2019]? If not, what are the exact differences?

**Limitations:**

The authors are clear on the limitations of the work.

---

> ### Author Rebuttal · Authors · 2023-08-06
>
> We thank the reviewer for taking the time to read the paper and for providing important feedback, which we will carefully consider and incorporate in the next revision.
>
> First of all, we confirm that studying SLSC can inform algorithm design. As we discuss in the last paragraph of page 5, given any two policies, our results provide us with a tool to generate a spectrum of policies that interpolate their values. This means that we can generate a continuum of optimal policies if we find two (possibly by gradient descent from different initializations). If the agent has a secondary preference over these policies (for example, some policies are easier to implement on the physical actuator), an eventually more preferred policy can be selected. In addition, in the paper [Jin 2022] pointed out by Reviewer efCX, the authors introduced a pseudo-linearity structure, which is weaker than and gets implied by our result in Theorem 1. This structure plays an important role in their algorithm design process, which strengthens our belief that the results on connected superlevel sets can inspire and guide the design of future algorithms that we may not foresee at this point.
>
> In the broad minimax optimization and game theory literature, knowledge on the existence of Nash equilibrium (NE) guides the design of algorithms and helps researchers understand the limit of any algorithms that can be designed. In nonconvex-nonconcave minimax optimization problems, global NE may not always exists, and weaker notions of optimality have been introduced including local NE [Daskalakis 2018, Mazumdar 2018], coarse correlated equilibria [Muller 2022, Mao 2023], and local/global min-max equilibria [Jin 2020, Vamvoudakis 2023]. Algorithms that search for these alternative solutions are designed by exploiting their specific structure, which may not be optimal in the NE sense even if the existence of NE is established later on.
>
> As we discussed in the related works section, our analysis and network architecture in the deep RL setting are inspired by [Nguyen 2019], which studies the optimization landscape of a supervised learning problem with a convex objective function. Assumptions on the piecewise linearity and monotonicity of the activation functions are required in [Nguyen 2019]. As our objective is a non-convex value function and the last layer of our neural network has to use a nonlinear, non-monotone softmax activation function to produce a valid probability distribution, important innovations need to be made to handle the activation function and the interfacing between the neural network and the policy optimization objective. The analysis of the first and last layer of our neural networks especially reflect the innovation.
>
>
>
> References
>
> Daskalakis, Constantinos, and Ioannis Panageas. "The limit points of (optimistic) gradient descent in min-max optimization." Advances in neural information processing systems (2018).
>
> Mazumdar, Eric, Lillian J. Ratliff, and S. Sastry. "On the convergence of gradient-based learning in continuous games." arXiv preprint arXiv:1804.05464 (2018).
>
> Nguyen, Quynh. "On connected sublevel sets in deep learning." In International conference on machine learning, pp. 4790-4799. PMLR, 2019.
>
> Jin, Chi, Praneeth Netrapalli, and Michael Jordan. "What is local optimality in nonconvex-nonconcave minimax optimization?." In International conference on machine learning, pp. 4880-4889. PMLR, 2020.
>
> Muller, Paul, Romuald Elie, Mark Rowland, Mathieu Lauriere, Julien Perolat, Sarah Perrin, Matthieu Geist, Georgios Piliouras, Olivier Pietquin, and Karl Tuyls. "Learning Correlated Equilibria in Mean-Field Games." arXiv preprint arXiv:2208.10138 (2022).
>
> Jin, Yujia, Vidya Muthukumar, and Aaron Sidford. "The complexity of infinite-horizon general-sum stochastic games." arXiv preprint arXiv:2204.04186 (2022).
>
> Mao, Weichao, and Tamer Başar. "Provably efficient reinforcement learning in decentralized general-sum markov games." Dynamic Games and Applications 13, no. 1 (2023): 165-186.
>
> Vamvoudakis, Kyriakos G., Filippos Fotiadis, João P. Hespanha, Raphael Chinchilla, Guosong Yang, Mushuang Liu, Jeff S. Shamma, and Lacra Pavel. "Game theory for autonomy: From min-max optimization to equilibrium and bounded rationality learning." In 2023 American Control Conference (ACC), pp. 4363-4380. IEEE, 2023.

---

### Official Review · Reviewer_efCX · 2023-08-05

**Soundness:** 3 good
**Presentation:** 3 good
**Contribution:** 3 good
**Rating:** 5
**Confidence:** 3

**Summary:**

This work studies the connectedness in (deep) reinforcement learning. First, the authors show that the superlevel set of average reward objective in reinforcement learning is connected under both tabular and over-parameterized policies. The objective is shown to satisfy a stronger equiconnectedness property. Second, the authors use the results to get minimax theorems for robust reinforcement learning. In particular, they show that show that minimax problems with convex functions on one side and equiconnected functions on the other side observes the minimax equality (i.e. has a Nash equilibrium).

**Strengths:**

The superlevel set of average reward objective in reinforcement learning is connected seems a novel and interesting result.

The results holds for over-parameterized neural networks, and find applications in minimax problems.

**Weaknesses:**

Some simulations could be used to verify the results.



**Questions:**

1. The authors showed that gradient dominance and connectedness do not imply one another. This raises the question of whether connectedness properties of objective functions allow any algorithms to find optimal solutions?

2. Related question: It seems that in Figure 1, the left example is easy for optimization (separated global maximizers), while the right example seems to be harder for optimization (stationary points which are not global maximizer). Does this imply that comparing to gradient dominance, connectedness is a less favourable property for optimization?

3. There are existing results showing that in game settings, a property called pseudo-linearity is satisfied, i.e., there are monotonically increasing path between two policies, see Theorem 1 of https://arxiv.org/abs/2204.04186. I am curious if there is any relation between the connectedness in this work and pseudo-linearity in the above paper?

**Limitations:**

This is a mostly theoretical work. There is no negative societal impact.

---

> ### Author Rebuttal · Authors · 2023-08-06
>
> We thank the reviewer for bringing to our attention this highly relevant pseudo-linearity structure. From the way we construct the path map in the proof of Theorem 1, it is not difficult to see that our result implies the pseudo-linearity, but pseudo-linearity does not imply connected superlevel sets as it lacks the sense of connectedness. The pseudo-linearity structure plays an important role in the algorithm design process in the paper [Jin 2022], which strengthens our belief that the results on connected superlevel sets can inspire and guide the design of future algorithms that we may not foresee at this point.
>
> Besides its potential application to algorithm design, our result in Section 4 discusses something more fundamental. The connectedness of superlevel sets allows us to derive the existence of a globally optimal solution (global Nash equilibrium), which in general may not exist for a nonconvex-concave minimax optimization problem. Knowing that the solution exists is a prerequisite before any algorithms can be designed to find the solution. When the existence of Nash equilibrium is unclear, we usually need to compromise by considering weaker notions of optimality. Please see our response to Reviewer ceyJ for a short list of alternative local/global optimality notions that have been proposed in nonconvex-concave and nonconvex-nonconcave minimax optimization.
>
> References
>
> Jin, Yujia, Vidya Muthukumar, and Aaron Sidford. "The complexity of infinite-horizon general-sum stochastic games." arXiv preprint arXiv:2204.04186 (2022).

---

### Decision · Program_Chairs · 2023-09-21

**Decision:**

Accept (poster)

**Comment:**

This paper studies the optimization landscape of average reward objective in reinforcement learning. The main results are the superlevel set of the objective with respect to policy parameter is connected under direct and over-parameterized neural network parameterizations, found to be interesting by all reviewers. The authors then used the findings to get minimax theorems for robust reinforcement learning.